# Antimicrobial Stewardship in COVID-19 Patients: Those Who Sow Will Reap Even through Hard Times

**DOI:** 10.3390/antibiotics12061009

**Published:** 2023-06-04

**Authors:** Marcella Sibani, Lorenzo Maria Canziani, Chiara Tonolli, Maddalena Armellini, Elena Carrara, Fulvia Mazzaferri, Michela Conti, Annarita Mazzariol, Claudio Micheletto, Andrea Dalbeni, Domenico Girelli, Evelina Tacconelli

**Affiliations:** 1Infectious Diseases Department, Azienda Ospedaliera Universitaria Integrata Verona, 37126 Verona, Italy; 2Division of Infectious Diseases, Department of Diagnostics and Public Health, University of Verona, 37129 Verona, Italy; 3Department of Pharmacy, Azienda Ospedaliera Universitaria Integrata Verona, 37126 Verona, Italy; 4Microbiology and Virology Section, Department of Diagnostic and Public Health, University of Verona, 37129 Verona, Italy; 5Respiratory Unit, Cardio-Thoracic Department, Azienda Ospedaliera Universitaria Integrata Verona, 37126 Verona, Italy; 6Section General Medicine C and Liver Unit, Department of Medicine, Azienda Ospedaliera Universitaria Integrata Verona, 37126 Verona, Italy; 7Department of Medicine, Section of Internal Medicine D, University of Verona, 37129 Verona, Italy

**Keywords:** COVID-19, antimicrobial stewardship, antibiotic consumption

## Abstract

Background: Since the SARS-CoV-2 pandemic emerged, antimicrobial stewardship (AS) activities need to be diverted into COVID-19 management. Methods: In order to assess the impact of COVID-19 on AS activities, we analyzed changes in antibiotic consumption in moderate-to-severe COVID-19 patients admitted to four units in a tertiary-care hospital across three COVID-19 waves. The AS program was introduced at the hospital in 2018. During the first wave, COVID-19 forced the complete withdrawal of hospital AS activities. In the second wave, antibiotic guidance calibration for COVID-19 patients was implemented in all units, with enhanced stewardship activities in Units 1, 2, and 3 (intervention units). In a controlled before and after study, antimicrobial usage during the three waves of the COVID-19 pandemic was compared to the 12-month prepandemic unit (Unit 4 acted as the control). Antibiotic consumption data were analyzed as the overall consumption, stratified by the World Health Organization AWaRe classification, and expressed as defined-daily-dose (DDD) and days-of-therapy (DOT) per 1000 patient-day (PD). Results: In the first wave, the overall normalized DOT in units 2–4 significantly exceeded the 2019 level (2019: 587 DOT/1000 PD ± 42.6; Unit 2: 836 ± 77.1; Unit 3: 684 ± 122.3; Unit 4: 872, ± 162.6; *p* < 0.05). After the introduction of AS activities, consumption decreased in the intervention units to a significantly lower level when compared to 2019 (Unit 1: 498 DOT/1000 PD ± 49; Unit 2: 232 ± 95.7; Unit 3: 382 ± 96.9; *p* < 0.05). Antimicrobial stewardship activities resulted in a decreased amount of total antibiotic consumption over time and positively affected the watch class and piperacillin-tazobactam use in the involved units. Conclusions: During a pandemic, the implementation of calibrated AS activities represents a sound investment in avoiding inappropriate antibiotic therapy.

## 1. Introduction

The pandemic caused by the severe acute respiratory syndrome coronavirus 2 (SARS-CoV-2) has deeply impacted countless aspects of the national healthcare system. Among others, the inappropriate use of antimicrobial agents, especially during the first phase of the pandemic, raised special concern in terms of antibiotic stewardship (AS) [1] and the possible spread of multidrug-resistant (MDR) bacteria [2]. Several authors report an increase in antibiotic consumption, particularly during the first months of the COVID-19 pandemic, with respect to pre-COVID-19 times [3,4,5,6,7,8,9,10].

Although several guidance documents for antibiotic usage have been developed to recommend against routine usage of antibiotics in this population [11,12], the estimated proportion of COVID-19 patients receiving antibiotic therapy is close to 60% [1,13]. Several factors have been recognized as potential drivers of antibiotic overprescription: reduction of AS activities due to personnel reallocation, decreased screening for MDR organisms, shortage of specific antibiotics, difficulty in diagnosis of coinfections, and rapid turn-over of personnel [14,15,16]. However, there is evidence reporting a very low rate of bacterial coinfections in patients with SARS-CoV-2 infection. In a meta-analysis reviewing data up to April 2020, Langford et al. [17] and Lansbury et al. [18] found a rate ranging between 4 and 6% of patients and up to 14% of healthcare-associated infections in critically ill patients in intensive care units (ICU) [19]. Similarly, in a recent meta-analysis including studies up to May 2021, the prevalence of confirmed bacterial coinfection was 4% in the overall population and 12% in critically ill patients [1].

To tackle the misuse of antibiotics, various strategies have been proposed. However, clear recommendations on AS in a pandemic or in infectious diseases with pandemic potential have not been developed due to limited evidence [4,20,21,22,23].

Our work aims to substantially add to the existing evidence by evaluating the impact of a multiphase and customized AS intervention in non-ICU COVID-19 wards during the first three waves of the COVID-19 pandemic.

## 2. Results

Overall, the intervention included 1743 patients and 29,112 PD.

### 2.1. Antimicrobial Consumption

Nearly 40,000 individual drug administrations were analyzed. During the first pandemic wave (March–June 2020), overall consumption largely exceeded the desirable consumption estimate of 587 days of therapy (DOT)/1000 patient days (PD) (95% C.I. 559.4–613.7) (based on the levels of consumption achieved after prepandemic AS intervention [24]) for all the units (Unit 2: 836, 95% C.I. 143.0–1528; Unit 3: 684, 95% C.I. 489–878.1; Unit 4: 872, 95% C.I. 468.1–1275.9), but Unit 1, which was the last to be activated in April. Figure 1 shows the overall anatomical therapeutic chemical classification system (ATC) J01 antimicrobial consumption across the study period compared to the prepandemic consumption level.

After the intervention, consumption reduced in all the wards. Consumption in Units 1–3 significantly reduced compared to the 2019 level, while in Unit 4 overall consumption data fell in the referral range. The annual whole-hospital antimicrobial consumption expressed by defined daily dose (DDD)/1000 PD was of 715, 811, and 732 in 2019, 2020, and 2021, respectively. Table 1 summarized the mean overall consumption per wave and per unit, compared to the referral consumption level. 

Overall normalized antimicrobial consumption as expressed by DOT showed a significant and progressive decrease across the three waves for Unit 1 (−29 DOT/1000 PD, −5.5%), Unit 2 (−604 DOT/ 1000 PDs, −72%), and Unit 3 (−302 DOT/1000 PDs, −44%), while no significant variation emerged for Unit 4 (control Unit). For Units 2 and 3, significant reductions over time occurred also for DDD, length of therapy (LOT), and World Health Organization (WHO) watch class antimicrobials. Detailed data are provided in Table 2. Antibiotic consumption according to WHO AwaRe classes is shown in Figure 2.

When consumptions were stratified according to WHO AWaRe classes [25], we observed substantial variation between units for watch antimicrobials: Unit 4 showed higher consumption when compared to all the other wards in both waves two and three; the difference in the amount of employed piperacillin-tazobactam had a similar trend, accounting for 30–50% of the total watch variation. Amoxicillin-clavulanate was the most prescribed antibiotic from the access class in all four wards and in all periods, accounting for 68%, 82%, 49%, and 63% of total access consumption in Units 1, 2, 3, and 4, respectively. Considering watch class, piperacillin/tazobactam accounted for one-third of the consumption (32–44%), followed by ceftriaxone (20–27%) and meropenem (7–15%). In the reserve class, linezolid was the most commonly used agent (36–58%), followed by new cephalosporins/beta-lactamase inhibitors (11–32%) and daptomycin (9–31%).

### 2.2. Microbiological and Clinical Outcomes

Positive blood cultures were detected in 7% of patients and were stable over time for each unit (ranging 3–10% in individual wards). Multidrug-resistant bacteria were etiological agents in 2.7% of positive blood cultures, ranging from 1.1 to 4.0% according to the unit. C. difficile infections were stable over the waves and compared with the prepandemic period (23 cases in total, 0.8 cases/100 admitted patients, <1.5/1000 PD for every period analyzed). No clusters were detected. No significant difference in microbiological outcomes emerged when analyzed within or between the three units across time.

The mean mortality rate across all three periods was 16%; it was higher for Units 1 and 4. When analyzed over the three waves, no significant variation emerged for any ward. The mean length of stay (LOS) was 7 ± 2, 7 ± 1.5, and 8 ± 3 days in the three waves, respectively. No significant variation intra- or interunits could be identified.

### 2.3. Qualitative Indicators

During the third wave (February–May 2021), 22 prospective audits were conducted in the three units involved in AS activities, with 503 individual patient charts being reviewed. The prevalences of patients receiving any antibiotic therapy on the audit day in Units 1, 2, and 3 were, respectively, 37%, 22%, and 24%. The overall prescribing appropriateness ranging 67–74%. Targeted therapies accounted for 33–52% of the total prescription, with a mean appropriateness of 78%.

## 3. Discussion

Several authors reported increasing antibiotic consumption in the first months of the pandemic when compared to previous years and were early advocates of the AS principles being applied and promoted even in this difficult situation [21]. Despite this, very few AS studies were implemented in real-life COVID-19 patients. In this study, we showed that an AS intervention calibrated for COVID-19 patients can control the risk of increased inappropriate antibiotic therapy during a pandemic. Antimicrobial consumption in all wards peaked in wave 1, exceeding the pre-pandemic consumption level in Units 2–4. After AS intervention implementation, consumption tends to reduce, but significant variation across the waves was observed only for Units 1–3 involved in the enhanced AS intervention, where lower WHO watch antimicrobial consumption also occurred.

Published studies of antibiotic use during the COVID-19 pandemic mainly report aggregate whole-hospital normalized antibiotic consumption [9,23,26,27] rather than assessing specifically COVID-19 dedicated wards [28]. The whole-hospital consumption increase has been reported up to 10–15% when compared to the prepandemic period [26,28]; in our facility, we observed a +13% increase in the whole-hospital overall antibiotic consumption (DDD/1000 PD) between 2019 and 2020, while focusing on the COVID-19 wards in the first wave, the variation was between −10% and +48% (mean + 24%). Our reported level of absolute consumption in the first wave was close to the one of 700.3 (±354.8) DDD/1000 occupied bed days (OBD) reported by Guisado-Gil et al. in a tertiary-care hospital in Spain during the first COVID-19 wave [28]. Most of the studies also underline that the higher level of consumption observed during the first months of the pandemic [4,20,22] was followed by a sharp reduction after the immediate introduction of the simple AS bundle [22,29]. Y Liew et al. [20] recorded an increase in defined daily dose (DDD)/100 patient-days (PD) in the first months of 2020 vs. 2019 (54 vs. 47); nonetheless, by the third month into the pandemic, DDD/100 bed day gradually declined to settle at levels similar to the previous year. A. Murgadella-Sancho et al. [4] noted that the mean consumption of antibiotics during hospitalization was lower in 2020 than in 2019 (57.8 DDD/100 PD vs. 64.7 DDD/100 PD), except for March 2020 (80 DDD/100 PD). M. Staub et al. [22] observed weekly duration of therapy (DOT)/1000 PD in medical and ICU wards: the former experienced an increase of 145.3 DOT/1000 PD initially, followed by a decline (362 DOT/1000 PD) after implementation of a bundle of AS interventions; the latter experienced an initial rise of 204, then a reduction of 226.3 DOT/10000 PD. 

In our study, analyzing the decreasing consumption trend over subsequent COVID-19 waves, significant variation was identified only in the wards where enhanced AS activities were implemented. The comparisons of DOT and LOT levels and trends for each ward provide some useful insight: in both Units 2 and 3, the decrease in DOT between waves 1 and 2 largely overcomes the reduction in LOT, thus reflecting not only a reduction in duration of therapy or prevalence of patients receiving antibiotics but also a substantial reduction in combination therapies; the DOT to LOT differences of the last two waves are pretty much closer, suggesting the further reduction in this last phase resulted from a reduced start or duration of antibiotic course. In Unit 1, no significant reduction in LOT emerged, suggesting that the reduction of combination therapies played a major role in the overall reduction. The prevalence of patients receiving any antibiotic treatment as recorded by audits performed in Units 1–3 during the third wave was below 40%, thus substantially lower than reported in the first COVID-19 months [1,13] but in line with the literature focusing on later pandemic phases [30].

Different patients’ case-mix (especially in terms of clinical severity and comorbidities) and the level of care provided certainly strongly influenced the total consumption level observed in each ward as well as the composition in terms of AwaRe classes. Units 2 and 3 admitted patients with lower clinical severity (WHO 3–4 severity index); prescribers in these units were also the most trained in AS. Since the very beginning of wave 2, antibiotic use has dramatically dropped and tended to stabilize at a level lower than expected in the medical area. This low consumption probably reflects a judicious antibiotic use in the COVID-19 moderately severe patients’ population, where a very low rate of coinfection and hospital-acquired infections occurred. This was also confirmed by the high prescribing appropriateness registered by audits. Units 1 and 4, on the contrary, cared mostly for patients with a more severe presentation (WHO 5 severity index) and patients from the ICU after clinical improvement. A higher rate of consumption could be forecast for this setting. Interestingly, despite higher personnel and patient turn-over and bed capacity, Unit 1 showed a substantially lower overall, watch class, and piperacillin/tazobactam consumption when compared to Unit 4. Unit 1 total normalized DOT was significantly lower than expected in waves 2 and 3; piperacillin/tazobactam level was the lowest consumption in wave 1 and showed the lowest mean across the whole 12-month period, thus preventing a significant trend from emerging. In Unit 4, on the contrary, although decreasing, the piperacillin-tazobactam DOT level represents the highest among the four wards in all the waves, thus suggesting further room to curb wide-spectrum antibiotic overprescribing.

The significant reduction or stability of Watch antimicrobials in the context of reducing consumption led to a favorable shift in the AWaRe relative composition of consumption, as shown in Figure 2, with the access representing close to 30% and the watch not reaching 70% of the total consumption in units involved in AS initiatives. Reserve antimicrobial prescription, as for hospital policy, was restricted to infectious disease consultants, and limited to target treatment of MDR-caused infections or specific indications based on the Italian Medicine Agency requirement, thus, we did not regard them as a target for our AS initiative.

Even accounting for different patients’ case-mixes and other possible biases, an association between AS activities and improved antibiotic use, both in quantitative and qualitative terms, emerged from these considerations. Different baseline prescribing skills, restricted resource availability and limited time availability suggested personalized AS activities for each ward: a baseline, early, and diffuse intervention was aimed at increasing antibiotic guidelines usability through the dissemination of a mobile and web-based app; then periodical infectious disease (ID)-attendance to clinical rounds was introduced, prioritising resources based on the context complexity and previous AS training; finally, prospective audits were introduced to further focus on and ensure improvement not only in the amount but primarily in the quality of prescriptions.

Bloodstream infections were uncommon and stable over time; *C. difficile* infections were rare, and no clusters were detected. These observations appear to be in line with the literature [17,26]. COVID-19 mortality was deeply entwined with the epidemic phase, demographics, clinical presentation, and standard of care [31]. In our study, the crude in-hospital mortality rate varied widely with unit and wave, with the higher rate observed in units admitting more severe patients and providing higher intensity of care; overall mortality rate of 16%, which is in line with published reports for in-hospital mortality [32,33,34]. No data suggested that reduced antibiotic use was associated with increased mortality, thus confirming the safety of the intervention.

### Strengths and Limitations

There are several strengths to this study. Most importantly, our AS intervention consisted of simple and highly replicable actions: the introduction of internal guidelines, the attendance of clinical rounds by an ID specialist, and the use of prospective audits. Moreover, the process and the results of this intervention were evaluated using different indicators, belonging to different domains, such as appropriateness of description through audit, antimicrobial consumption summarized from prescription-level data, and microbiological and clinical outcomes. Data were systematically collected through the hospital data repository for the whole study period. Selected metrics of antibiotic consumption were robust within each other and showed similar levels of consumption to independent evaluations through audits. 

This study is not without limitations. The full-time allocation of 1 ID specialist represented the most resource-consuming aspect of this intervention, limiting its feasibility, especially in small hospitals. Specific to our setting, time-varying biases due to the pandemic’s continuously changing landscape limited the results’ generalizability. These biases may be represented, for example, by changes in clinical practice, the case mix of patients, and personnel turnover. In a rapidly evolving situation such as COVID-19, it remains difficult to measure treatment effects, especially in the AS setting. The inconstant activation of COVID-19 wards based on the extremely variable rate of hospitalization made data collection time points intermittent and disjointed, thus preventing us from performing interrupted time series analysis as generally recommended to evaluate AS initiative effectiveness. [35]. Microbiological outcomes were not tailored to the COVID-19 pandemic: samples from the respiratory tract could have represented a better estimate of the incidence of bacterial co-infection. The common pitfalls of AS studies that are present in our study are the use of surrogate measures (e.g., rate of positive blood cultures representing infection rate) and the use of aggregated data that limits the statistical approach.

## 4. Materials and Methods

A controlled before-and-after study was conducted in a 1350-bed tertiary care, university hospital in Verona, Italy, from March 2020 to May 2021. For the purpose of the study and data analysis, the COVID-19 pandemic was stratified in 3 waves: March–June 2020; October 2020–January 2021; and February–May 2021.

Antimicrobial consumption in the COVID-19 wards was first compared to pre-COVID-19 consumption data. Using the published data from the SAVE AS intervention [24], we assumed that the normalized antibiotic consumption captured in the AS intervention follow-up in 2019 would represent the desirable antimicrobial consumption for the medical wards in our hospital. 

The whole hospital’s annual antimicrobial consumption data from 2019–2021 was also calculated to identify general trends and provide a benchmark.

Then, we analyzed the antimicrobial consumption in the study wards across the three pandemic waves to evaluate whether any variation occurred before and after the implementation of an enhanced, COVID-19-calibrated AS intervention in 3 wards; another COVID-19 ward, not involved in the enhanced AS activities, served as a control.

### 4.1. Setting

The study includes 4 units reserved for COVID-19 patients not requiring mechanical ventilation: Unit 1 had a bed capacity ranging from 15 to nearly 50 and a 16-bed semi-intensive section for patients requiring noninvasive ventilation (NIV) or high-flow nasal cannula (HNFC) (WHO Ordinal Scale for clinical improvement level 5); Unit 2 admitted both severe and moderate patients and was used as a “step-down” ward for post-acute patients (WHO outcome scale 3–4) with a bed capacity increasing from 20 to 42; Unit 3 had 34-bed and provided standard care (low-flow oxygen to HNFC but no NIV), treating predominantly severe COVID-19 patients (WHO Severity score 4); and Unit 4 admitted subcritical and post-ICU patients with a bed capacity ranging from 10 to 20, selected as control unit. The period of activity and patient days (PD) for each ward across the three main pandemic waves occurring in our geographical area are shown in Figure 3.

### 4.2. Intervention

The University Hospital of Verona implemented an AS team in 2018. In the same year, a comprehensive AS intervention was implemented in 4 hospital medical wards and achieved a significant reduction in antimicrobial consumption sustained beyond the intervention’s completion in the 21-month post-intervention period. During 2019, consumption in the included wards stabilized with a mean value of 587 DOT/1000 PDs (95% C.I. 559.4–613.7). As extensive audits simultaneously detected high prescribing appropriateness, we assumed this level would represent a fair estimate of the desired consumption level in the specific context of the local medical area [24].

During the first COVID-19 wave, AS activities were interrupted as the AS teams were fully dedicated to COVID-19 management. Formal internal guidelines addressing COVID-19 treatment and antibiotic management in COVID-19 patients were disseminated in all 4 units at the end of the first wave. No antibiotic treatment was routinely recommended for COVID-19 patients, regardless of clinical severity. The guidelines limited antibiotic therapy for those patients presenting with clinical, laboratory, or radiological data, suggesting bacterial coinfection; in that case, referral to the already existing hospital guidelines for empiric antibiotic treatment was advised. ID consultations were available upon request for all the non-ID-led units.

In the second wave, AS interventions were progressively re-established and calibrated on the COVID-19 patients, with diversified enhanced activities in Units 1–3:Since October 2020, an ID specialist has attended daily Unit 1 clinical rounds to support antibiotic prescription and withholding and advise on the diagnostic process. Biweekly revision of ongoing antibiotic therapies was also resumed in Unit 2, already involved in the pre-COVID-19 AS intervention, to refresh physicians’ diagnostic and prescribing skills.Starting from the third wave, the COVID-19 guidelines as well as the hospital antibiotic guidelines were made available through the Firstline app, available for iOS and Android, and on the web (https://firstline.org/, accessed on 28 April 2023) for Units 1–3, to increase their usability; local epidemiological data and monographs of antibiotics were also accessible from the same platform.Prospective audits were conducted weekly in the three intervention units during the third wave. All the patients receiving antibiotic therapy on the audit day had their electronic health records reviewed; quality indicators such as compliance with empirical therapy guidelines and appropriateness of therapy were evaluated and recorded.

The intervention timeline is shown in Figure 3.

### 4.3. Outcomes

The primary outcome was the overall antibiotic consumption measured as defined daily dose (DDD), days of therapy (DOT), and length of therapy (LOT) and normalized per 1000 patient days (PD). Defined daily dose is defined as the assumed average maintenance dose per day for a drug used for its main indication in adults, which was calculated according to the WHO ATC Index [36]. DOT is defined as the aggregate sum of days for which any amount of a specific antimicrobial agent was administered to an individual patient (i.e., if a patient receives more than one antibiotic, more than one DOT per day would be counted), while LOT represents the number of days that a patient receives systemic antimicrobial agents, irrespective of the number of different antibiotics [37].

For the 4 units, included in the study, antimicrobial consumption data encompassing all the ATCJ01 drugs administered to patients were retrieved from the hospital’s electronic prescribing system. Whole-hospital consumption data were retrieved by the pharmacy’s annual report based on the drug dispensing database.

As secondary outcomes, we analyzed:consumption data broken down to a single agent and stratified by WHO AWaRe classification (access, watch, reserve) [25].prescribing appropriateness as registered by prospective audits. All the patients receiving antibiotic therapy on the audit day had their clinical charts reviewed for presumptive infective diagnosis, antimicrobial prescription, and microbiological results. Appropriateness of therapy was defined as compliance with antibiotic guidelines for empirical therapy and as adequate coverage plus de-escalation if needed for targeted, microbiological-based, therapy. The prevalence of patients receiving antibiotics on the audit day was also collected.Clinical outcomes, including in-hospital mortality (measured as crude rate in-hospital mortality) and length of stay (LOS).microbiological outcomes, including total bloodstream infection (BSI)*,* BSI caused by MDR bacteria (i.e., methicillin-resistant *S. aureus* and *S. epidermidis,* carbapenem-resistant Enterobacterales and *P. aeruginosa*, ESBL-producing gram-negative, vancomycin-resistant Enterococci), and *C. difficile* infections (incidence per 100 admitted patients, deduplication was applied, counting only the first isolates/positive tests per patient in a 28-day interval, common contaminants were manually removed).

All the outcomes were measured monthly.

### 4.4. Statistical Analysis

Descriptive statistics were used for antibiotic consumption.

A Student’s *t*-test was employed to compare the individual unit overall antimicrobial consumption in each of the three subsequent COVID-19 waves to the 2019 mean consumption. Variation in consumption within each ward across the three periods was appraised using an ANOVA for repeated measures. A *p*-value less than 0.05 was regarded as significant. The analysis was carried out using STATA software (© StataCorp LLC, College Station, TX, USA).

## 5. Conclusions

Even in a challenging context such as a pandemic, attentively allocating resources to retain AS programs in place represents a sound investment in order to preserve the quality of care and the patient’s safety. Essential enabling AS activities can be readapted to effectively face the emerging need even in a resource-constrained setting to ensure that the essential level of prescribing appropriateness is met.

## Figures and Tables

**Figure 1 antibiotics-12-01009-f001:**
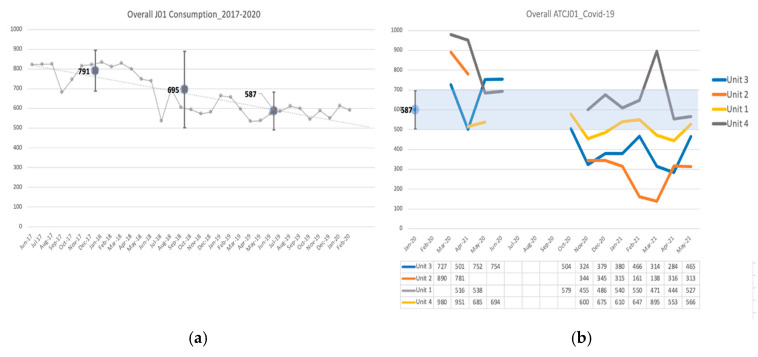
Comparison of the overall ATC-J01 antimicrobial consumption in the prepandemic period and during the COVID-19 pandemic: (**a**) Consumption trend (DOT/1000 PD) in the hospital’s medical area targeted by the hospital’s AS program in the period 2017–2019 [24]; (**b**) consumption trends (DOT/1000 PD) in the 4 COVID-19 dedicated wards; monthly consumption data in the COVID-19 period are provided in the table.

**Figure 2 antibiotics-12-01009-f002:**
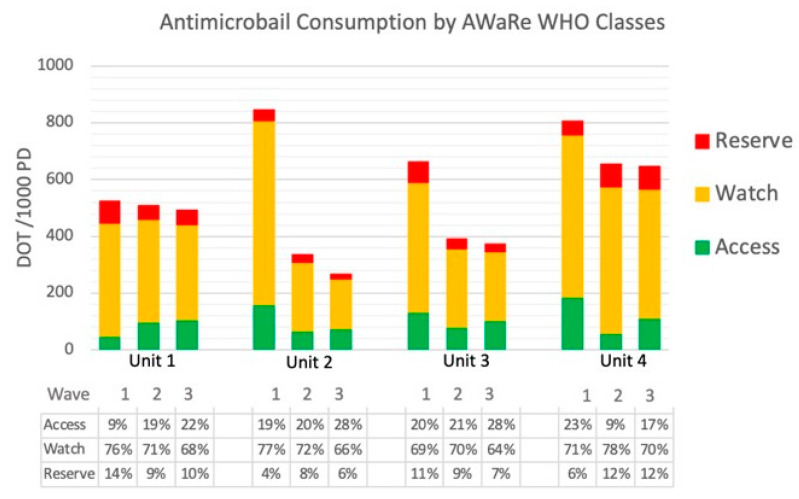
Normalized DOT/1000 PD of AWaRe antibiotics; overall consumption is reported as the percentage of access, watch, and reserve by COVID-19 waves.

**Figure 3 antibiotics-12-01009-f003:**
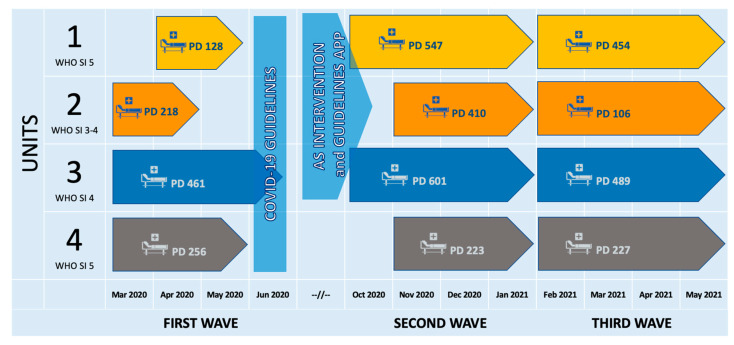
AS intervention timeline and patient days (PD) per unit per wave. *WHO SI = World Health Organization Ordinal Scale for clinical improvement in COVID-19 patients: 1: ambulatory patients, no limitation of activities; 2: ambulatory patients, with limitation of activities; 3: hospitalized patients, no oxygen therapy needed; 4: hospitalized patients, oxygen by mask or nasal cannulae needed; 5: hospitalized, severe disease, noninvasive ventilation, or high flow oxygen needed.

**Table 1 antibiotics-12-01009-t001:** Comparison of the overall antimicrobial consumption (DOT/1000 PD) to the desirable consumption estimate for the Medical Area [21].

Antimicrobial Consumption (ATC J01) DOT/1000 PD							
Unit	Medical Area Desirable Consumption Estimate Mean (SD)	Wave 1 Mean (SD)	*p*-Value *	Wave 2 Mean (SD)	*p*-Value *	Wave 3 Mean (SD)	*p*-Value *
Unit 1 (WHO Scale 5)		527 (±15.6)	0.082	515 (±55.3)	**0.0168**	498 (±49)	**0.0037**
Unit 2 (WHO Scale 3–4)	587 (±42.6)	836 (±77.1)	**<0.001**	335 (±17.0)	**<0.001**	232 (±95.7)	**<0.001**
Unit 3 (WHO Scale 4)		684 (±122.3)	**0.027**	397 (±76.1)	**<0.001**	382 (±96.9)	**<0.001**
Unit 4 (C) (WHO Scale 4–5)		872 (±162.6)	**<0.0001**	628 (±40.7)	0.1496	665 (±159)	0.1236

* Student’s T test; C = control. WHO Scale = World Health Organization Ordinal Scale for clinical improvement in COVID-19 patients: 1: ambulatory patients, no limitation of activities; 2: ambulatory patients, with limitation of activities; 3: hospitalized patients, no oxygen therapy needed; 4: hospitalized patients, oxygen by mask or nasal cannulae needed; 5: hospitalized, severe disease, noninvasive ventilation, or high flow oxygen needed.

**Table 2 antibiotics-12-01009-t002:** Comparison of antimicrobial consumption data across waves and units.

Outcome	Unit	First Wave Mean (DS)	Second Wave Mean (DS)	Third Wave Mean (DS)	*p*-Value *
DOT/1000 PD	**Unit 1**	527 (±15.6)	515 (±55.3)	498 (±49)	**<0.05**
	**Unit 2**	836 (±77.1)	334.7 (±17.0)	232 (±95.7)	**<0.05**
	**Unit 3**	684 (±122.3)	397 (±76.1)	382 (±96.9)	**<0.05**
	Unit 4(C)	872 (±162.6)	628 (±40.7)	665 (±159)	>0.05
DDD/1000 PD	Unit 1	635 (±217.1)	575 (±94.2)	533 (±80.9)	>0.05
	**Unit 2**	913 (±137.9)	319(±34)	219 (±96.3)	**<0.05**
	**Unit 3**	736 (±150.4)	408 (±67.9)	430 (±111.7)	**<0.05**
	Unit 4(C)	834 (±209.8)	576 (±42.7)	636 (±183.0)	data
LOT/1000 PD	Unit 1	407 (±43.8)	444 (±49.0)	411(±38.6)	>0.05
	**Unit 2**	614 (±36.1)	294 (±15)	201 (±74.4)	**<0.05**
	**Unit 3**	524 (±73.5)	327 (±55.1)	307 (±69.7)	**<0.05**
	Unit 4(C)	702 (±95.6)	532 (±50.7)	514 (±102.8)	data
ACCESS	Unit 1	57 (±26.2)	101 (±23.6)	107.5 (±8.2)	>0.05
(DOT/1000 PDs)	Unit 2	157 (±31.8)	67 (±16.1)	52 (±47.9)	>0.05
	Unit 3	155 (±91.7)	84 (±35.1)	109 (±34.7)	>0.05
	Unit 4(C)	194 (±71.4)	60 (±29.5)	111 (±41.3)	>0.05
WATCH	Unit 1	369(±101.8)	379 (±61.5)	338.8 (±33.9)	>0.05
(DOT/1000 PDs)	**Unit 2**	640 (±72.2)	243 (±26.9)	172 (67.7)	**<0.05**
	**Unit 3**	456 (±83)	277 (±38.8)	245 (56.5)	**<0.05**
	Unit 4(C)	632 (±103.3)	513 (±116.6)	472 (82.3)	>0.05
RESERVE	Unit 1	101 (±91.2)	35 (±35.1)	52 (±13.9)	>0.05
(DOT/1000 PDs)	Unit 2	39 (±26.9)	25 (±14.4)	9 (±10.3)	>0.05
	Unit 3	72 (±9.9)	36 (±12.3)	29 (±22.0)	>0.05
	Unit 4(C)	46 (±22.9)	56 (±70.4)	81 (49.3)	>0.05
PIPERACILLIN-TAZOBACTAM	Unit 1	93 (±55.2)	112 (±37.7)	143 (±49.6)	>0.05
(DOT/1000 PDs)	Unit 2	161 (±43.8)	124 (±15.5)	73.2 (±16.4)	**<0.05**
	Unit 3	142 (±63.6)	114 (±23.6)	103 (±29.9)	**<0.05**
	Unit 4(C)	276 (±32.8)	225 (±34.6)	212 (±39.0)	**<0.05**

* ANOVA test for repeated measures; C = control.

## Data Availability

The data presented in this study and not contained within the article are available upon request from the corresponding author. The data are not publicly available due to hospital ethical policies.

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
