# Peer review of "Antimicrobial Stewardship in COVID-19 Patients: Those Who Sow Will Reap Even through Hard Times"

_antibiotics, 2023, doi:10.3390/antibiotics12061009_

Round 1
Reviewer 1 Report
This manuscript examines the outcomes of implementing an AS program during different phases of the CV19 pandemic. The data are consistent with the authors' conclusions and show the effectiveness of the strategy in a resource-limited setting during a pandemic. The paper could be significantly strengthened with a more thorough description of proposed AS interventions - share specific strategies that were used in these wards, how appropriate those may be in other settings, and what a typical strategy may be that administrators could consider using.
Generally good - a few places where tense (past/present) is confused. Other small things, such as on line 64 "AS stewardship" is redundant and should just be "AS".
Author Response
We are thankful to the reviewer for the precious suggestion and the time spent assessing the manuscript.
Point 1: The paper could be significantly strengthened with a more thorough description of proposed AS interventions - share specific strategies that were used in these wards, how appropriate those may be in other settings, and what a typical strategy may be that administrators could consider using.
Response 1: We really appreciate this suggestion and the opportunity to describe the specific interventions in each Unit more thoroughly, we extend section 4.2 in order to do so. We also expand the discussion paragraph focusing on the different AS activities, calibrated for each setting.
Point 2: a few places where tense (past/present) is confused.
Response 2: We have an extensive English language revision and we fixed the tenses where needed.
Point 3: Other small things, such as on line 64 "AS stewardship" is redundant and should just be "AS".
Response 3: We remove the redundant word "stewardship" from line 64.
Reviewer 2 Report
Thank you for your submission.
For table 1. Please add a severity description on Unit 1, Unit 2, Unit 3, Unit 4 in the footer so readers can identify were sicker patients reside.
If available, it would be helpful to the readers to know which antibiotics overall were most commonly used in each category: reserve, watch, and access.
Line 131-138. With regard to overall prescribing appropriateness. Please add a line on whether that changed or not between study periods.
In Table 2. with regard to piperacillin-tazobactam (TZP) utilization, please include a discussion on why Unit 1 with Antimicrobial stewardship intervention had no change in (TZP) consumption. And yet in Unit 4 without AS intervention, the consumption of TZP decrease significantly after wave 1. Alternatively, you can mention something in your study limitation about other confounds that may have attributed to the utilization trend for TZP and unit 1 & 4 such as the severity of illness was not measured.
Figure 3: add description in footer of WHO SI 3-4 & 5
In your discussion, you need to expand on the limitations of the AS program. The AS program intervention did not impact reserve or access agents's DOT/1000PD. The bulk of the impact appears to be on Watch Agents with less severely ill patients in Unit 2 and 3.
Overall, good paper!
minor issues.
Author Response
We thank the reviewer for the valuable suggestion and the opportunity to improve the clarity of the manuscript
Point 1: For table 1. Please add a severity description on Unit 1, Unit 2, Unit 3, Unit 4 in the footer so readers can identify where sicker patients reside.
Response 1: Thank you for the suggestion, we added the severity description in the footer.
Point 2: If available, it would be helpful to the readers to know which antibiotics overall were most commonly used in each category: reserve, watch, and access.
Response 2: We really appreciate the reviewer’s insightful suggestion so we added a brief paragraph at the end of section 2.1 detailing the most employed agents for each category across Units (lines 124-130).
Point 3: Line 131-138. With regard to overall prescribing appropriateness. Please add a line on whether that changed or not between study periods.
Response 3: We agree with the reviewer that describing appropriateness variation across waves would be interesting as it represents one of the most relevant outcomes for stewardship activities; Unfortunately, due to resource limitation and stepwise implementation of stewardship activities, prospective audits assessing prescribing appropriateness were performed only during the third wave, so we don't have appropriateness data from the previous waves for comparisons. We extend the intervention description in the Method section detailing stewardship activities carried out during each wave.
Point 4: In Table 2. with regard to piperacillin-tazobactam (TZP) utilization, please include a discussion on why Unit 1 with Antimicrobial stewardship intervention had no change in (TZP) consumption. And yet in Unit 4 without AS intervention, the consumption of TZP decrease significantly after wave 1. Alternatively, you can mention something in your study limitation about other confounds that may have attributed to the utilization trend for TZP and unit 1 & 4 such as the severity of illness was not measured.
Response 4: We agree with the reviewer that this represents a striking founding. As discussed in lines 217-222, Unit 1 showed the absolute lowest TZP consumption in both wave 1 as well as the lowest TZP consumption if considering the whole period (March 2020-May 2021); On the contrary, Unit 4 TZP consumption was the highest considering both the above-mentioned periods. Moreover, although showing a significantly decreasing trend, the lowest value in Unit 4 during the third wave is close to 150% of the value collected in Unit 1 in the same period. The two Units cared for similar patients in terms of severity of illness and level of care, and Unit 1 showed high appropriateness when prescriptions were evaluated through audits: we believe that constant low consumption level prevented a significant trend to emerge and we are not sure that a further reduction would be desirable, as that low level is probably close to the appropriate consumption in that setting.
Point 5: Figure 3: add description in footer of WHO SI 3-4 & 5
Response 5: Thank you, we added the description as suggested.
Point 6: In your discussion, you need to expand on the limitations of the AS program. The AS program intervention did not impact reserve or access agents's DOT/1000PD. The bulk of the impact appears to be on Watch Agents with less severely ill patients in Unit 2 and 3.
Response 6: We agree with the reviewer that a reduction in absolute consumption of Access and Reserve groups antimicrobials was not observed, but we are not conceived this represents a failure or a limitation for an AS program.
AWaRe Classification was proposed as a tool to improve monitoring and defining goals for stewardship activities, but these latter needs to be calibrated to the specific setting. Moreover, the WHO 13th General Programme of Work 2019-2023 also regards an increase in the rate of Access antimicrobials to at least 60% of the total as a desirable target at a Country level.
We believe that the primary goal of stewardship is improving prescribing appropriateness and patient safety, while variation in consumption could be used only as a proxy outcome. Pursuing always a reduction in consumption, especially when considering specific classes/agents rather than the overall consumption, is not always feasible not desirable. In our study setting, in the context of an overall reduction of consumption, an increase or stability in the absolute consumption of Access agents results in a favorable shift in the relative composition, towards prescribing narrower spectrum agents. We added a line, to explain that in our hospital, Reserve antimicrobials prescription is restricted to Infectious disease specialists, and limited to target treatment or specific indication, according to the National Medicine Agency, thus, as MDR-caused infection rate didn't show major variation across time, we didn't regard them as a target for our AS initiative.
Reviewer 3 Report
I highly salute the approach of preserving anti-infective drugs, even in times of pandemic. Although this paper describes one center where protocol of antibiotics prescription was strict, this is a great example of ethical management in regards of future generation of patients treated with these drugs.
My suggestions to Authors are only technical>
1. Please make sure that all abbreviations are defined at first appearance, abstract of paper included.
2. No sentences should start with abbreviations or numbers.
Author Response
We thank the reviewer for taking the time to assess our manuscript.
As suggested we checked and amend the manuscript assuring all definitions were adequately provided and no sentences would start with abbreviation or numbers.